# The Transcription of Flight Energy Metabolism Enzymes Declined with Aging While Enzyme Activity Increased in the Long-Distance Migratory Moth, *Spodoptera frugiperda*

**DOI:** 10.3390/insects13100936

**Published:** 2022-10-16

**Authors:** Yan Fu, Ting Wu, Hong Yu, Jin Xu, Jun-Zhong Zhang, Da-Ying Fu, Hui Ye

**Affiliations:** 1Yunnan Academy of Biodiversity, School of Biodiversity Conservation, Southwest Forestry University, Kunming 650224, China; 2Key Laboratory for Forest Resources Conservation and Utilization in the Southwest Mountains of China, Ministry of Education, Southwest Forestry University, Kunming 650224, China; 3School of Ecology and Environment, Yunnan University, Kunming 650091, China

**Keywords:** *Spodoptera frugiperda*, RNAseq, flight muscle, aging, energy metabolism, enzyme activity

## Abstract

**Simple Summary:**

In this study, we established the overall transcriptome framework of the thorax and aging-associated expression change trajectories, and constructed the possible flight energy metabolism pathways using potential energy metabolic enzymes in *Spodoptera frugiperda* females. Gene expression and functional prediction indicated that most genes and pathways related to energy metabolism and other biological processes (e.g., transport, longevity and signaling) were downregulated with aging, whereas the expression of enzymes of proline oxidation and genes related to immunity and repair increased with age. The increased expression of proline oxidation enzymes with age suggests that this energy metabolic pathway also is important or linked to some aging processes. Further activity assay showed that the activities of the five tested key metabolic enzymes increased with age. The transcriptional decrease but activity increase with aging imply that these enzymes are stable. A high activity of enzymes may be crucial for a long-distance migrator. The activity ratio of GAPDH/HOAD was <1 and decreased with age, indicating that lipid is the main flight fuel for this insect, particularly in aged individuals.

**Abstract:**

Of all the things that can fly, the flight mechanisms of insects are possibly the least understood. By using RNAseq, we studied the aging-associated gene expression changes in the thorax of *Spodoptera frugiperda* females. Three possible flight energy metabolism pathways were constructed based on 32 key metabolic enzymes found in *S. frugiperda*. Differential expression analysis revealed up to 2000 DEGs within old females versus young ones. Expression and GO and KEGG enrichment analyses indicated that most genes and pathways related to energy metabolism and other biological processes, such as transport, redox, longevity and signaling pathway, were downregulated with aging. However, activity assay showed that the activities of all the five tested key enzymes increased with age. The age-associated transcriptional decrease and activity increase in these enzymes suggest that these enzymes are stable. *S. frugiperda* is a long-distance migrator, and a high activity of enzymes may be important to guarantee a high flight capacity. The activity ratio of GAPDH/HOAD ranged from 0.594 to 0.412, suggesting that lipid is the main fuel of this species, particularly in old individuals. Moreover, the expression of enzymes in the proline oxidation pathway increased with age, suggesting that this energy metabolic pathway also is important for this species or linked to some aging-specific processes. In addition, the expression of immunity- and repair-related genes also increased with age. This study established the overall transcriptome framework of the flight muscle and aging-associated expression change trajectories in an insect for the first time.

## 1. Introduction

Insects are the most successful and diverse group of organisms on the earth. The ability to fly is crucial for their survival and reproduction, which evolved relatively quickly across the class [1,2]. Insect flight is driven by flight muscles that are composed mainly of large myofibril that are histologically different from the common skeletal muscle fibers [3]. Myofibril consists of thick and thin filaments. The thick filament consists primarily of the motor molecule myosin, which interacts with the thin filament’s actin molecules and functions in the sliding motion between the two sets of filaments. Other conserved myofibril proteins were also found in the flight muscle, such as tropomyosin, troponin and titin. Later studies also found some flight muscle-specific proteins, such as flightin, troponin-H and glutathione S-transferase [4]. Flightin can bind to myosin rods to maintain the integrity and stability of sarcomeres [5]. Glutathione S-transferase interacts with troponin-H in the thin filament [6], which may be essential for the mechanics of the flight muscle while their precise function is still poorly known.

Insect flight is a very complicated and extremely energy-demanding process. The frequency of wing beat can be very high (up to 1000 Hz) in insects and thus metabolic activity is very high during flight [7]. The evolution of fuel use by insect flight muscles is still poorly understood. So far, the energy metabolisms of only a few species have been subjected to detailed examination [8]. Insects may use carbohydrates (primarily trehalose), fats (mostly diacylglycerol) and proline to fuel flight, which can be different between species and between stages of flight [8]. In some long-distance migratory insects, such as some moths and locusts, the flight is initially fueled by carbohydrates, and a long-distance flight is accompanied by the activation of fatty acid oxidation [9]. Bees mainly use carbohydrate to fuel flight [8] while some beetles use proline as the principal flight fuel [10]. Accordingly, some key enzymes that function in these energy metabolisms during flight have been identified in these insects, such as the trehalase that catalyzes the conversion of trehalose to glucose, glyceraldehyde-3-phosphate dehydrogenase and phosphoglycerate kinase that catalyze glyceraldehyde 3-phosphate to pyruvate [9]. However, our knowledge on the mechanisms of insect flight and related energy metabolism is still limited.

Muscle aging inevitably results in the dysfunction, reduced performance and mass of muscles, which is a complex, multifactorial process [11,12]. As an important problem in biology, the underlying processes and mechanisms of muscle senescence have attracted the interest of researchers from pharmacology, physiology, ecology and evolution [11,12,13]. Quite a number of studies in insects have shown that aging profoundly impairs flight muscle structure [14], energy metabolism [15] and flight capacity [14,16]. For example, in the oriental fruit fly, *Bactrocera dorsalis*, the myofibrillar diameter and the size of mitochondria, as well as flight capacity decreased in aged females [14]. Studies in mammals suggested that mitochondrial dysfunction appears to have a key role in age-related decline of muscle function [17]. In the blowfly, *Sarcophaga bullata*, pyruvate-proline respiration in flight muscle mitochondria declined in senescent insects [18].

The growing sophistication of gene expression technologies and bioinformatics has led to the idea that aging can be understood by analyzing the transcriptome. Transcriptome-based mapping trajectories of gene expression changes in different tissues of different organisms have provided deep insights into how biological functions change with age [19,20]. Studies in mammal skeletal muscle transcriptome have established a detailed framework of the global transcriptome and mRNA isoforms that govern muscle damage and homeostasis with age [21,22]. For example, specific mRNAs that changed significantly with age in human skeletal muscles were enriched for several aging-related processes, including cellular senescence, insulin signaling, myogenesis, oxidative phosphorylation and adipogenesis [21]. However, the aging of insect flight muscles has not been studied by using transcriptome. Further research based on these findings and high-throughput technologies will help to deepen our understanding of insect biochemistry and evolutionary biology, and may also help to answer questions relevant to human health [9].

The fall armyworm, *Spodoptera frugiperda*, is a major transnational migratory pest that is currently occurring in the world [23,24]. This pest was first discovered in Yunnan Province of China at the end of 2018, and then quickly spread northward to the vast areas of China [25]. The ability of long-distance migration is the key determinant of invasion status and evolutionary success of *S. frugiperda* [26]. Our previous study had revealed the flight capacity and flight muscle structure of *S. frugiperda* females [27], which provide a good basis for our in-depth research in this field. Moreover, the aging-related transcriptional changes and enzyme activity in the flight muscle of *S. frugiperda* females have not been investigated. In the present study, therefore, we studied the overall transcriptome framework of the thorax and aging-associated changes, and constructed the possible but the most detailed flight energy metabolism pathways using all potential key energy metabolic enzymes in *S. frugiperda* females. We further tested the effect of age on the activity of five key energy metabolic enzymes.

## 2. Materials and Methods

### 2.1. Insects and Sampling

The larvae of *S. frugiperda* were collected in corn field that near Dongchuan town in Yunnan Province, China. The larvae then were reared on the artificial diet [28] under 28 ± 1 °C and 60–80% relative humidity with 14:10 h light:dark photoperiod. Adults were fed with a 10% honey solution. Their offspring was used for the present study.

To ensure virginity and age, male and female pupae were sexed based on morphological characteristics [29] and then were caged separately. Newly eclosed female adults were collected and reared in plastic boxes (fed with 10% honey solution) for the following treatment and sampling. Females were sampled at 1-, 3-, 4-, 7- and 10-days-old. The adult lifespan of this insect is about 11 d under above rearing condition (Fu Y., unpubl. data). All insects were sampled at midnight (6 h into the scotophase). Thoraxes were cut from the insects and wings, feet and scales were removed from the thorax. For transcriptome, 10 thoraxes (from 10 insects) were used as replicates, and 3 replicates were used for each age. For enzyme activity assay, 8 thoraxes (from 8 insects) were used as replicates, and 4 replicates were used for each age. All samples were frozen in liquid nitrogen immediately after sampling and stored at −80 °C.

### 2.2. cDNA Library Preparation and Sequencing

The Trizol reagent (Invitrogen Inc., Calsbad, CA, USA) was used to isolate total RNA from samples following the manufacturer’s instructions. RNA was further purified by using the RNeasy MinElute Clean up Kit (Qiagen Inc., Valencia, CA, USA) and eluted in RNA Storage Solution (Ambion Inc., Austin, TX, USA). RNA integrity was verified on an Agilent Bioanalyzer 2100 system (Agilent Technologies Inc., Palo Alto, CA, USA). The sequencing cDNA libraries were prepared by using NEBnext Ultra RNA Library Prep Kit for Illumina (New England BioLabs Inc., San Diego, CA, USA) following the manufacturer’s protocols. The libraries were sequenced by Illumina Hiseq4000 platform (Illumina Inc., San Diego, CA, USA) and the paired-end reads were generated.

### 2.3. Quality Control and Assembly

The obtained raw reads were processed by trimming the adapter and low-quality reads to produce clean reads. Clean reads were then assembled using Trinity v2.5.1 to generate transcripts [30]. Transcript redundancies were removed using TGICL software v2.1 to acquire unigenes without redundancy [31]. The clean reads were then mapped to the reference genome sequence of *S. frugiperda* (https://www.ncbi.nlm.nih.gov/genome/?term=spodoptera%20frugiperda) (accessed on 21 December 2021) using Hisat2 v2.0.5 software.

### 2.4. Differential Expression Analysis and Enrichment Analysis of Differentially Expressed Genes (DEGs)

Transcripts per million (TPM) was used to determine gene expression levels and the edgeR R package (version 3.0.8) was used to analyze the differential expression between treatments. The significance threshold of *p*-value in multiple tests was adjusted by *q*-value [32]. We used *q* < 0.05 and |log2(foldchange)|>1 as the threshold to judge the significance of gene expression differences.

In order to well understand the biological function of DEGs, they were fully annotated based on the following databases: NR, Pfam, Swiss-Prot, Gene Ontology (GO), KEGG Orthology (KO) and Cluster of Orthologous Groups (COG).

GOSeq program was used to perform GO enrichment analysis and KOBAS software was used to implement KEGG enrichment analysis of DEGs. GO terms and KEGG pathways with *q* < 0.05 were significantly enriched in DEGs. These enriched terms and pathways were grouped based on their functions. In order to establish an overall picture of age-related transcriptome changes, all genes (DEGs and non-DEGs) which fall into these groups were identified and their expression change trajectories with age were analyzed and illustrated.

Aging is related to protein synthesis. We thus further analyzed the expression changes of the eukaryotic translation initiation factor 4E (eIF4E) with age, which is a key mRNA translation initiation factor. Significant differences in the expression of this gene between different ages were analyzed using an ANOVA.

### 2.5. Validation of RNAseq Sequencing Data by Using Real-Time Quantitative PCR

Real-time qPCR was performed to verify the accuracy of the RNAseq data. Six enzyme DEGs [LOC118268804 (ENO), LOC118281561 (PK), LOC118268149 (GPDH), LOC118271716 (GAPDH), LOC118263360 (THI), and LOC118278275 (ACDH); please see their full names and related pathways in Table 1] were selected for qRT-PCR analysis in different age comparison groups. The *Actin* (GeneBank ID: LOC118279073) was used as a reference gene. *Actin* has been regarded as a housekeeping gene and has been widely used as a reference gene in insects. In this study, the expression of this gene did not show significant changes between samples of *S. frugiperda*. RNAiso plus (TaKaRa Inc., Dalian, China) was used to isolate total RNA from samples and PrimeScript RT reagent Kit (Takara Inc., Dalian, China) was used to generate cDNA for qPCR. The PCR was carried out on a QuantStudio 7 Flex System (Thermo Fisher Scientific Inc., Waltham, MA, USA) with gene specific primers (Appendix A). The reactions condition is 95 °C for 30 s, followed by 40 cycles of 95 °C for 5 s, 60 °C for 30 s and dissociation. The specificity of the SYBR Green PCR signal was confirmed by melting curve analysis. The 2^−ΔΔCT^ method [33] was used to calculate the relative expression. Each experiment was repeated three times using three independent RNA samples. 

### 2.6. Effect of Age on Enzyme Activities

Activities of the 5 key metabolic enzymes (GPDH, GAPDH, LDH, HOAD and CS; please see their full names and related pathways in Table 1) in the thorax of *S. frugiperda* were assayed by available ELISA kits (Jiangsu Meimian industrial Co., Ltd., China; catalogue no. of MM-91327O1 for GPDH, MM-91324O1 for GAPDH, MM-925259O1 for LDH, MM-91249O1 for HOAD, and MM-91334O1 for CS) according to the manufacture’s protocol. Briefly, above collected samples for enzyme activities assay were homogenized using a tissue grinder in PBS (PH7.4; containing multiple protease inhibitors) solution at a ratio of 0.1 g sample to 0.9 mL of PBS. The homogenate was centrifuged at 3000 rpm for 20 min at 4 °C. The supernatant was collected for subsequent analysis. Sample or standard (50 μL) was added to the micro-ELISA plate wells which pre-coated with specific antibody for a specific enzyme. Shake the plate gently to allow the enzyme in the sample to combine with the antibody. Then, the enzyme labeling reagent (100 μL) was added to each well, and incubated for 60 min in water bath at 37 °C. Free components in the wells were washed away by using a wash buffer solution. Additionally, then the coloration reagent (100 μL) was added to each well, and incubated for 15 min in water bath at 37 °C. Stop solution (50 μL) was added to the wells to stop the coloration reaction. The optical densities (OD) value of each well of all samples was measured at 450 nm using a spectrophotometer. The OD value is proportional to the concentration of the enzyme in the sample. A standard curve of OD versus the enzyme concentration was produced using the calibration standards provided in the kit. The activities of the enzyme in the samples were determined by comparing to the OD value of the standard curve. 

### 2.7. Statistical Analyses

Significant differences in the expression of target genes between treatments measured by qPCR were analyzed by an ANOVA. Significant differences in the activity of different enzymes were analyzed by an ANOVA followed by Tukey’s studentized range (HSD) test for multiple comparisons. Linear regression between enzyme activity and age were analyzed using an analysis of regression (AOR).

## 3. Results

### 3.1. Summary of Sequencing Quality

RNAseq obtained ~55,000,000 clean reads from each of the 15 sequenced libraries, with Q20, Q30 and mapped ratios as 98.18–98.38%, 94.33–94.84% and 81.32–84.63%, respectively (Appendix A). Pearson’s correlation coefficient showed higher correlations between biological replicates (Appendix A) and principal component analysis (PCA) showed biological replicates cluster together (Appendix A), affirming the reproducibility of RNASeq and biological replicates. The RNAseq raw reads were deposited into the NCBI SRA database (BioProject ID: PRJNA857921).

### 3.2. Transcriptome-Based Discovery of Genes Encoding Flight Energy Metabolic Enzymes and Flight Muscle Structural Proteins

Based on known knowledge in insects and other organisms, 38 key enzymes were determined for flight energy metabolism in insects (Table 1). Blast searching indicated that most (32) of these enzymes exist in the transcripts of *S. frugiperda* (Figure 1; Appendix A). Three possible flight energy metabolism pathways were constructed based on these enzymes for *S. frugiperda* (Figure 1). The first pathway is the glycolysis pathway and the initial fuel of this pathway can be glycogen in the flight muscle and trehalose from the fat body, which involved 14 key enzymes. The second pathway is the fat oxidation pathway that involved seven enzymes, the initial fuel of which is diacylglycerol from the body fat. The third pathway is the proline oxidation pathway that involved four enzymes. All the three pathways were connected to the tricarboxylic acid (TCA) cycle, which involved seven enzymes.

A total of nine types of flight muscle structural proteins were identified from transcripts of *S. frugiperda*, with myosin and actin showing higher number of transcripts, with 26 and 41, respectively (Table 2).

### 3.3. Overview of Aging-Associated Transcriptional Changes

Differential expression analysis revealed 758 (462 down, 296 up), 1154 (707 down, 447 up), 2100 (1106 down, 994 up) and 1935 (1174 down, 761 up) DEGs within 3-day-old females versus 1-day-old females (D3:D1), D4:D1, D7:D1 and D10:D1 groups, respectively (Figure 2, Appendix A). There are 499 common DEGS within D3:D1 and D4:D1 groups, 535 common DEGS within D3:D1 and D7:D1 groups, 453 common DEGS within D3:D1 and D10:D1 groups, and 339 common DEGs among the four groups (Figure 3, Appendix A).

All these DEGs were annotated based on the six databases as mentioned above (Appendix A) and were enriched to GO terms (Appendix A) and KEGG pathways (Appendix A). These DEGs were mainly enriched to metabolism, defense, soma maintenance and aging-related GO terms (Appendix A) or KEGG pathways (Appendix A). These enriched GO terms and KEGG pathways were categorized into 15 groups based on their functions (Figure 4).

### 3.4. Aging-Associated Expressional Changes in Flight Energy Metabolic Enzymes and Flight Muscle Structural Proteins

All or most of the enriched terms (Figure 4a–d) or pathways (Figure 4e–h) related to energy metabolism, including carbohydrate metabolism, lipid metabolism, amino acid metabolism and TCA cycle were enriched to downregulated DEGs.

Correspondingly, with the increase in age, the expression of most metabolic enzymes in the glycolysis pathway (Figure 5, Appendix A), fat oxidation pathway (Figure 6, Appendix A) and TCA cycle (Figure 7, Appendix A) generally showed a downward trend. However, the expression of enzymes in the proline oxidation pathway showed an increased pattern with age, with some enzymes showing increased expression in middle-aged females and others showing increased expression in old females (Figure 8, Appendix A).

**Figure 4 insects-13-00936-f004:**
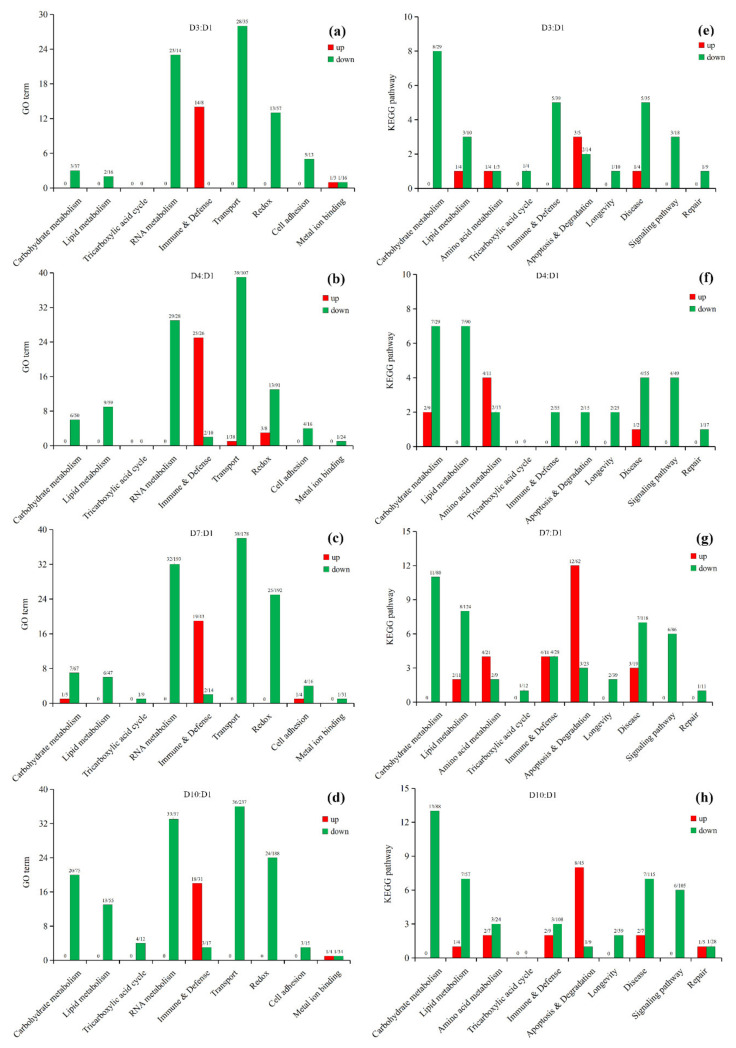
Enriched GO terms and KEGG pathways from the flight muscle of *S. frugiperda* females. (**a**–**d**) represent GO terms of D3:D1, D4:D1, D7:D1 and D10:D1 groups, respectively; (**e**–**h**) represent the KEGG pathways of D3:D1, D4:D1, D7:D1 and D10:D1 groups, respectively. Red columns represent terms or pathways enriched to upregulated DEGs and green columns represent terms or pathways enriched to downregulated DEGs. Upon the columns, the first number represents the number of terms or pathways, and the second number represents the number of DEGs.

The expression of flight muscle structural protein also declined with age. Among them, LOC118266930 (Troponin C) showed the highest expression (Figure 9, Appendix A).

### 3.5. Aging-Associated Expressional Changes in Other Gene Groups

All or most of the enriched terms (Figure 4a–d) or pathways (Figure 4e–h) related to RNA metabolism, transport, redox, cell adhesion, metal–ion binding, longevity, disease, signaling pathway and repair (including DNA and protein repair) were enriched to downregulated DEGs, while most of the enriched terms or pathways related to apoptosis and degradation, immune and defense were enriched to upregulated DEGs.

The expression of all transcripts (including DEGs and non-DEGs) related to RNA metabolism (Appendix A), transport (Appendix A), redox (Appendix A), longevity (Appendix A), disease (Appendix A), and signaling pathway (Appendix A) generally showed the same trend, i.e., decreased with the increase in age. For example, LOC118277946 (ATP synthase subunit b) from RNA metabolism, LOC118266866 (calcium-transporting ATPase) from transport, and LOC118266311 (protein lethal 2) from longevity showed high expression in young females and their expression declined quickly or gradually with age.

However, the expression of transcripts related to cell adhesion (Appendix A), metal–ion binding (Appendix A) and apoptosis and degradation (Appendix A) did not show a consistent change pattern with age. Such as LOC118280713 (vinculin-like) from cell adhesion, and LOC118271756 (sarcoplasmic calcium-binding protein 1) from metal–ion binding decreased with age. While LOC118265563 (integrin alpha-PS3) from cell adhesion and LOC118277523 (caspase-1-like) from apoptosis and degradation increased with age.

Moreover, the expression of transcripts related to repair (Appendix A) increased with aging, such as LOC118267122 (UV excision repair protein) and LOC118281733 (general transcription and DNA repair factor). Transcripts related to immune and defense (Appendix A) increased with age and showed a sharp increase in aged (D7) females and then declined, such as LOC118265131 (lysozyme-like), LOC118273606 (defense protein Hdd11) and LOC118279240 (attacin-like).

### 3.6. Validation of RNAseq Results by qPCR

Six enzyme DEGs were selected and their expression changes with age were verified by qPCR, where three, five, five, and five of these DEGs were used for D3:D1, D4:D1, D7:D1 and D10:D1 groups, respectively. The results showed that the expression levels of these genes (Figure 10) were similar to the RNAseq analysis results, indicating that the RNAseq data are reliable.

### 3.7. Effect of Aging on the Activity of Flight Energy Metabolic Enzymes

From an overall view, enzyme activity assay revealed remarkable difference on the activity of different enzymes (*F*_4,95_ = 273.66, *p* < 0.0001; Figure 11, Appendix A). Post-hoc multiple comparisons showed that HOAD had the highest activity (*p* < 0.05), followed by GPDH and GAPDH (*p* < 0.05), and then CS (*p* < 0.05), and LDH showed the lowest activity (*p* < 0.05) (Figure 11).

From a daily and lifetime perspective, all the five enzymes showed a rising trend in their activity, with HOAD, GPDH and LDH showing a temporary reduction at 4-day-old, and GAPDH and CS showing a temporary reduction at day 3 and day 4 (Figure 11). 

Analysis of regression showed that the activity of HOAD (*F*_1,18_ = 21.31, *p* < 0.0001), GPDH (*F*_1,18_ = 25.05, *p* < 0.0001), GAPDH (*F*_1,18_ = 7.05, *p* = 0.016) and LDH (*F*_1,18_ = 119.57, *p* < 0.0001) increased with age significantly (Figure 12), while CS did not show such trend significantly (*F*_1,18_ = 0.92, *p* = 0.349).

The ratio of GAPDH and HOAD activities were also measured, which can express the flight fuel use type in the energy metabolism of insects, with the ratio equal to or close to 1 using both lipid and carbohydrates, <1 mainly using lipid and >1 mainly using carbohydrates. The activity ratio of GAPDH:HOAD ranged from 0.4 to 0.6 in the thorax of *S. frugiperda* at different ages (Table 3), indicating that lipid is the main flight fuel in this insect.

## 4. Discussion

In this study, transcriptome and annotation identified nine families of flight muscle structural proteins in *S. frugiperda*, with myosin and actin showing higher number of transcripts (Table 2). Myosin is a superfamily of molecular motors that move along the track of actin filaments, which accounts for 60% of the total protein of myofibrils and plays a role in muscle contraction [63]. One flightin-encoding gene was also found in *S. frugiperda*. Flightin is a myosin-binding protein, which can bind to myosin rods and maintains the integrity of sarcomere [5]. Flightin orthologs were found not only in winged insects, but also in non-winged insects, and even non-insect species [66]. Flightin is a protein with deep ancestry and functions outside of flight muscles [66].

Insects in flight achieve the highest rates of metabolism known in all animals, but how they do this is still poorly known [2]. Studies suggested that the activity of most enzymes involved in fuel metabolism is much higher in insect flight muscle than in vertebrate skeletal muscle, and some enzymes in insect flight muscle have a high affinity for the myofibril and may exist as part of a complex that assists in the local generation of ATP [67]. A study in *Drosophila* showed GPDH isoform 1 (GPDH-1) is flight muscle-specific isoform and has a specific C-terminal tripeptide (QNL) that is indispensable for the localization of GPDH to the myofibril [68]. In the present study, we found two GPDH isoforms (Appendix A) in *S. frugiperda*, while we did not find the same or similar C-terminal tripeptide as showed in *Drosophila*. Database blast and reference searching also did not find such C-terminal tripeptide in insect species other than *Drosophila*, suggesting that the evolution of GPDH may be species specific. Interestingly, we found a conserved C-terminal tetrapeptide (PKCS) in *S. frugiperda* and other three lepidopterans, *S. litura*, *H. zea* and *M. sexta* (Appendix A). Whether this tetrapeptide play function in localization of GPDH to the myofibril is still unknown and worth further study.

Three possible flight energy metabolism pathways were constructed based on the identified 32 key enzymes for *S. frugiperda* (Figure 1; Table 1). All the three pathways were connected to the TCA cycle. The expression on enzyme-encoding genes related to the glycolysis pathway, fat oxidation pathway, and TCA cycle generally showed a downward trend with the increase in age (Figure 5, Figure 6 and Figure 7, Appendix A), which may be due to aging. However, the expression of enzymes in the proline oxidation pathway increased with age (Figure 8, Appendix A). Lepidoptera and Orthoptera species usually use carbohydrate at the initiation of flight, whereas lipid is the main fuel during sustained flight [7,9]. In several dipteran, orthopteran, lepidopteran and hymenopteran species proline is considered to provide intermediates (such as α-ketoglutarate; Figure 1) of the citric acid cycle, which is necessary for the enhanced oxidation of acetyl-CoA from a flight-induced increase in degradation of carbohydrate or lipid [7]. This may explain why the expression of enzymes in the proline oxidation pathway increased with age in *S. frugiperda*.

The activity of all the five tested enzymes (HOAD, GPDH, GAPDH, CS and LDH) increased with age in *S. frugiperda* (Figure 11), which is contrary to the result found in *Bactrocera dorsalis* [15]. *S. frugiperda* is a long distance migrator, younger adults have higher flight capacity, with 3-day-olds showing the highest flight capacity, while old (11-day-old) females still have strong flight capacity [27]. A higher activity of enzymes in aged adults may be important to guarantee a high flight capacity. The age-associated decrease in the expression of these enzyme-encoding genes and the increase on the activity of these enzymes may suggest that these enzymes are stable in *S. frugiperda*. The ratio of GAPDH and HOAD activities can express the flight fuel use type in the energy metabolism of insects [7]. In *Apis mellifica* (Hymenoptera) the activity ratio of GAPDH/HOAD is more than 1000, while the ratio is about 3 for *Locusta migratoria* (Orthoptera) and *Pieris brassicae* (Lepidoptera), and 0.1 for *Actias selene* (Lepidoptera) [69]. Later studies further showed that biotic (such as sex and aging) [15] and abiotic (such as green light) [70] factors may affect this ratio. In *Bactrocera dorsalis* (Diptera), the ratio is 14.66 and 18.08 for 1-day-old males and females, respectively; which decreased with the increase in age, and decreased to 1.0 in 25-day-old males and females [15]. In *S. exigua*, the ratio is 0.6 in mature pupae, which increase to 0.9 in 0-day-old adults and peaked (1.2) in 1-day-old adults, and then decreased to 0.9 in 7-day-old adults [71]. In the present study, the activity ratio of GAPDH/HOAD in *S. frugiperda* ranged from 0.594 to 0.412, and showed some decrease with age (Table 3), suggesting that lipid is the main fuel of this species, particularly in old individuals.

Human skeletal muscle transcriptome revealed 1134 aging-induced DEGs [21]. In the present study, the number of DEGs in aged moths compared to young ones increased with aging, from about 700 (D3:D1) to 2000 (D10:D1) (Figure 2). In addition to genes and pathways related to flight energy metabolic enzymes and flight muscle structural proteins as mentioned above, most other genes related to RNA metabolism, transport, redox, longevity, disease, and signaling pathway decreased with the increase in age. Muscle transcriptome in sheep showed genes related to muscle protein, phosphorylation, acetylation, metal binding and transport changed remarkably with aging, such as COX3, TPM2, MYL2, CYTB, ND4, TNNC and MYHT, their expression decreased with aging [22]. Homologous genes in *S. frugiperda*, such as AOB78_gp08 (COX3), AOB78_gp05 (ND4), AOB78_gp02 (CYTB), AOB78_gp12 (COX1), AOB78_gp11 (COX2), AOB78_gp04 (ND4L) and AOB78_gp09 (ATP6), also showed decreased expression with age (Appendix A). In addition, aging is related to protein synthesis. The eIF4E is a key mRNA translation initiation factor [72]. A deletion mutant of eIF4E resulted in decreased protein synthesis in *Caenorhabditis elegans* [73]. In this study, eIF4E expression level decreased significantly in aged females (Appendix A), representing that protein synthesis declined with age.

However, the expression levels of genes related to both repair and immune and defense increased with age. Age-dependent transcriptome changes in the human brain include decreased synaptic function and increased immune function [74]. Prolonged accumulation of cellular defects often results in immune activation [75]. There is sufficient evidence that pathways of DNA and protein repair become less efficient with aging, while much less is known about the causes of this deterioration [76]. In the present study, however, the increased expression trends on genes and pathways related to repair in old females may suggest an increased function, which needs further investigation.

## Figures and Tables

**Figure 1 insects-13-00936-f001:**
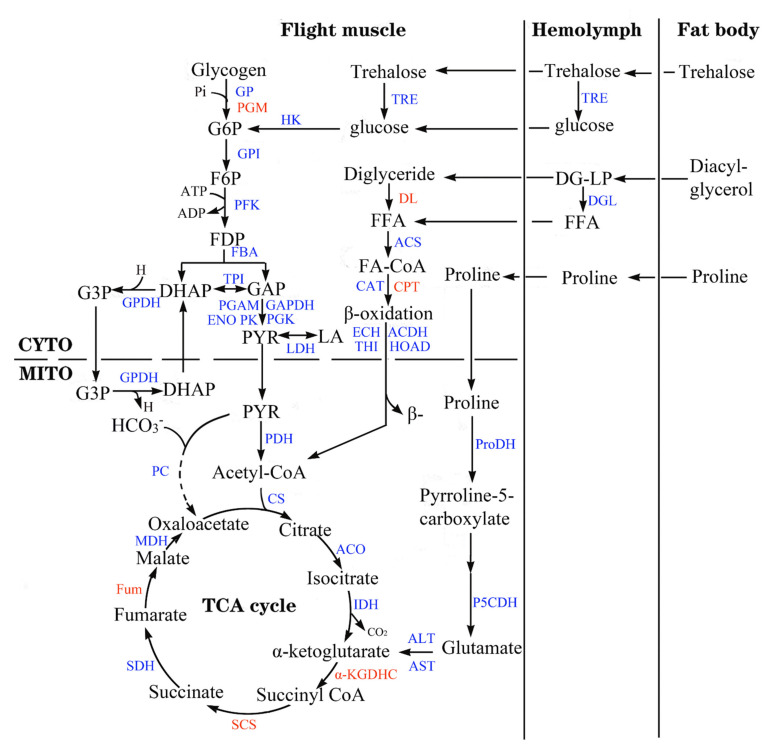
Possible flight energy metabolism pathways and key enzymes in *S. frugiperda*. The pathways with metabolic substrates and metabolites were constructed based on Thompson and Suarez [8]. Key enzymes were added to the pathways accordingly based on known knowledge in insects and other organisms. Black fonts represent substrates and metabolites, blue fonts represent enzymes found in *S. frugiperda*, and red fonts represent enzymes did not found in *S. frugiperda*. Full names of substrates, metabolites and enzymes can be found in Table 1.

**Figure 2 insects-13-00936-f002:**
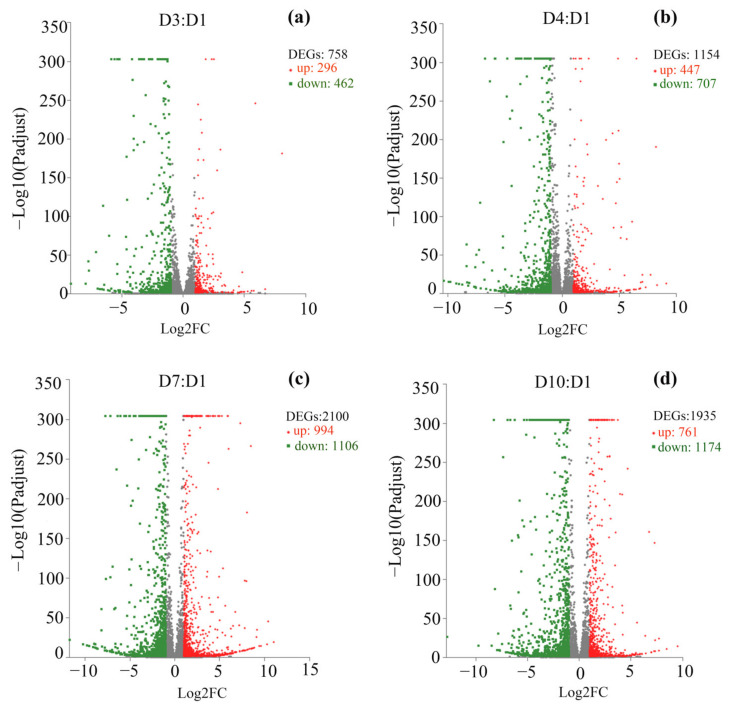
Volcano plots of DEGs from the flight muscle of *S. frugiperda* females. (**a**–**d**) are DEGs from the D3:D1, D4:D1, D7:D1 and D10:D1 groups, respectively. Genes with significant differential expression were indicated by red dots (upregulated) and green dots (downregulated). Genes with no significant differential expression were represented by gray dots.

**Figure 3 insects-13-00936-f003:**
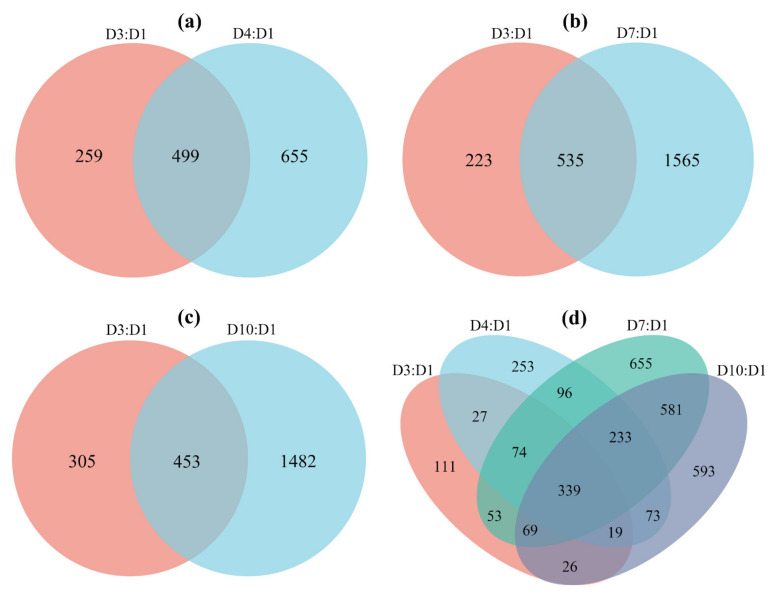
Venn diagrams of DEGs from the flight muscle of *S. frugiperda* females. (**a**–**c**) are common DEGs within D3:D1 and D4:D1 groups, D3:D1 and D7:D1 groups, and D3:D1 and D10:D1 groups, respectively; and (**d**) is common DEGs among the four groups (D3:D1, D4:D1, D7:D1 and D10:D1). The overlapping circles represented common DEGs among all combinations.

**Figure 5 insects-13-00936-f005:**
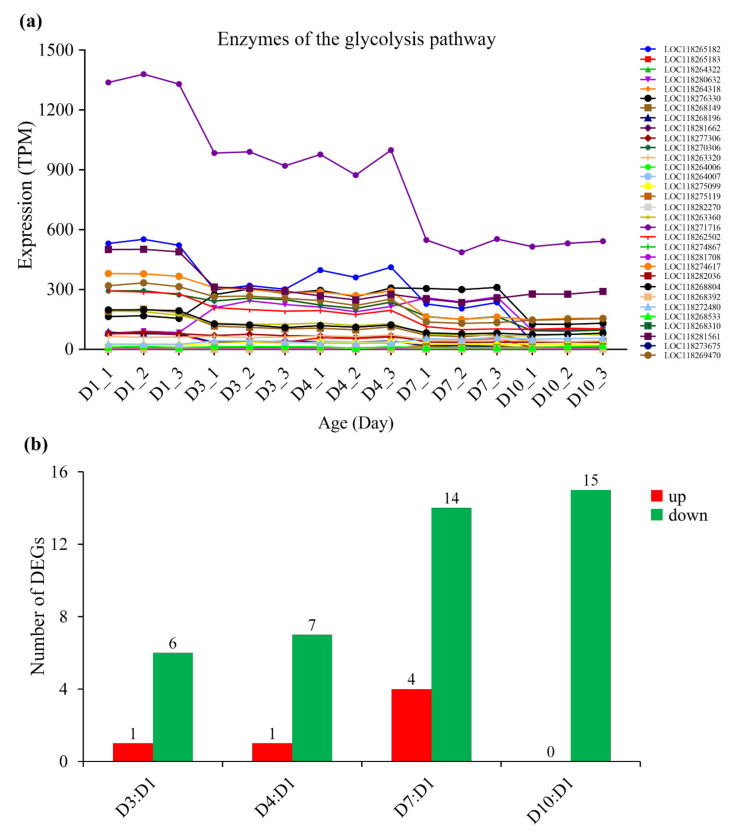
The expression of enzymes of the glycolysis pathway in *S. frugiperda* females at different ages. (**a**) Changes in enzyme gene expression with age; (**b**) DEGs of D3:D1, D4:D1, D7:D1 and D10:D1 groups, wherein red columns represent upregulated DEGs and green columns represent downregulated DEGs.

**Figure 6 insects-13-00936-f006:**
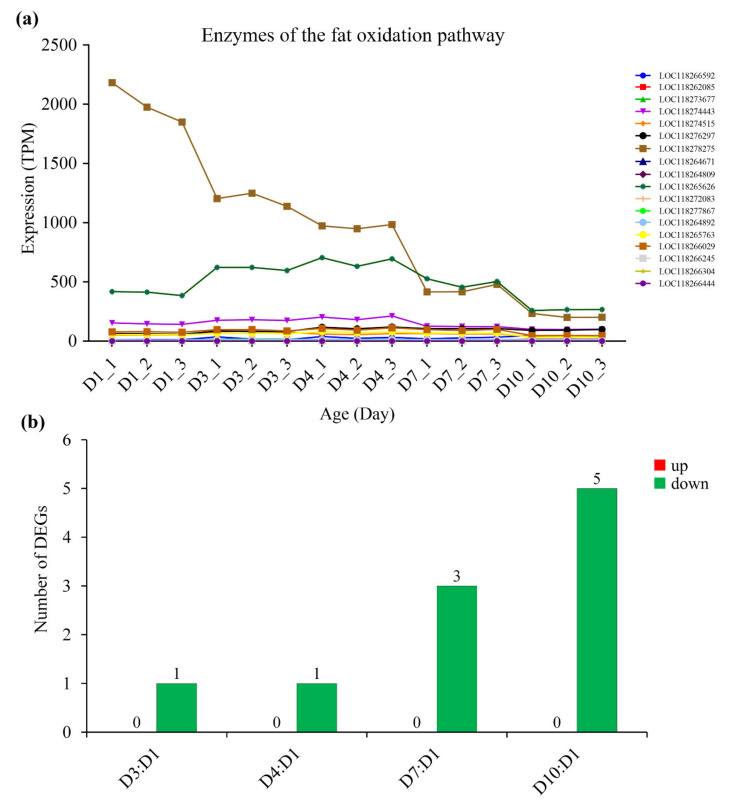
The expression of enzymes of the fat oxidation pathway in *S. frugiperda* females at different ages. (**a**) Changes in enzyme gene expression with age; (**b**) DEGs of D3:D1, D4:D1, D7:D1 and D10:D1 groups, wherein red columns represent upregulated DEGs and green columns represent downregulated DEGs.

**Figure 7 insects-13-00936-f007:**
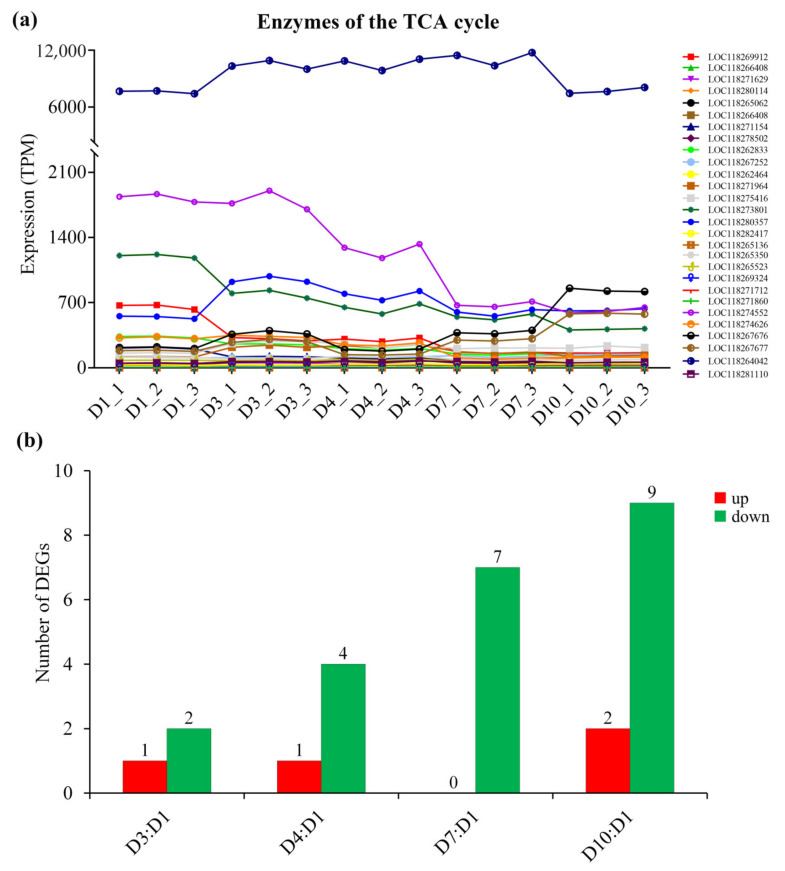
The expression of enzymes of the tricarboxylic acid (TCA) cycle in *S. frugiperda* females at different ages. (**a**) Changes in enzyme gene expression with age; (**b**) DEGs of D3:D1, D4:D1, D7:D1 and D10:D1 groups, wherein red columns represent upregulated DEGs and green columns represent downregulated DEGs.

**Figure 8 insects-13-00936-f008:**
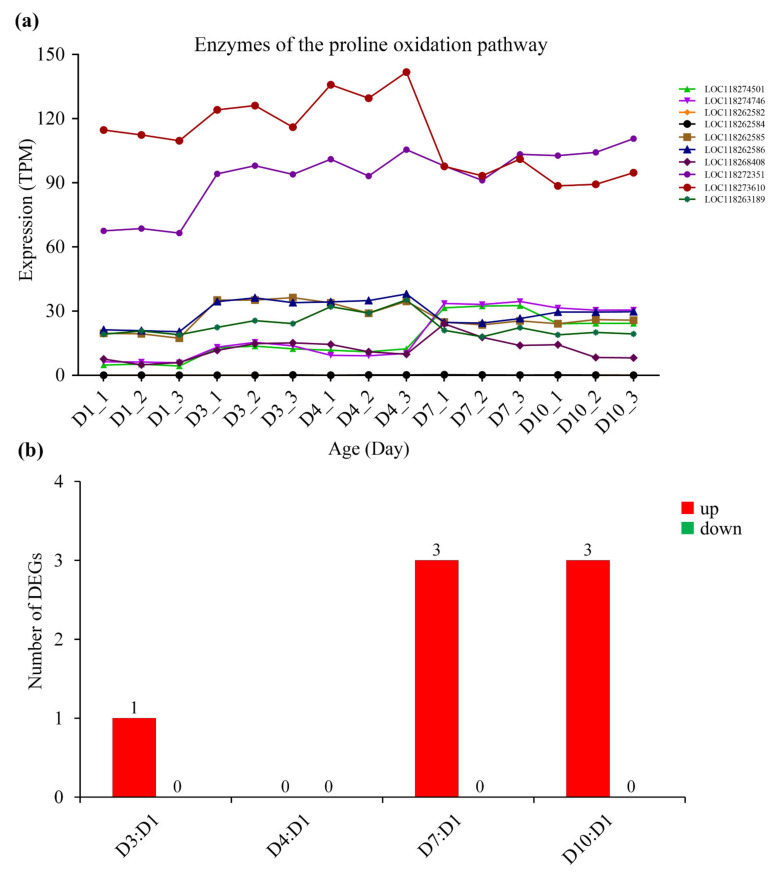
The expression of enzymes of the proline oxidation pathway in *S. frugiperda* females at different ages. (**a**) Changes in enzyme gene expression with age; (**b**) DEGs of D3:D1, D4:D1, D7:D1 and D10:D1 groups, wherein red columns represent upregulated DEGs and green columns represent downregulated DEGs.

**Figure 9 insects-13-00936-f009:**
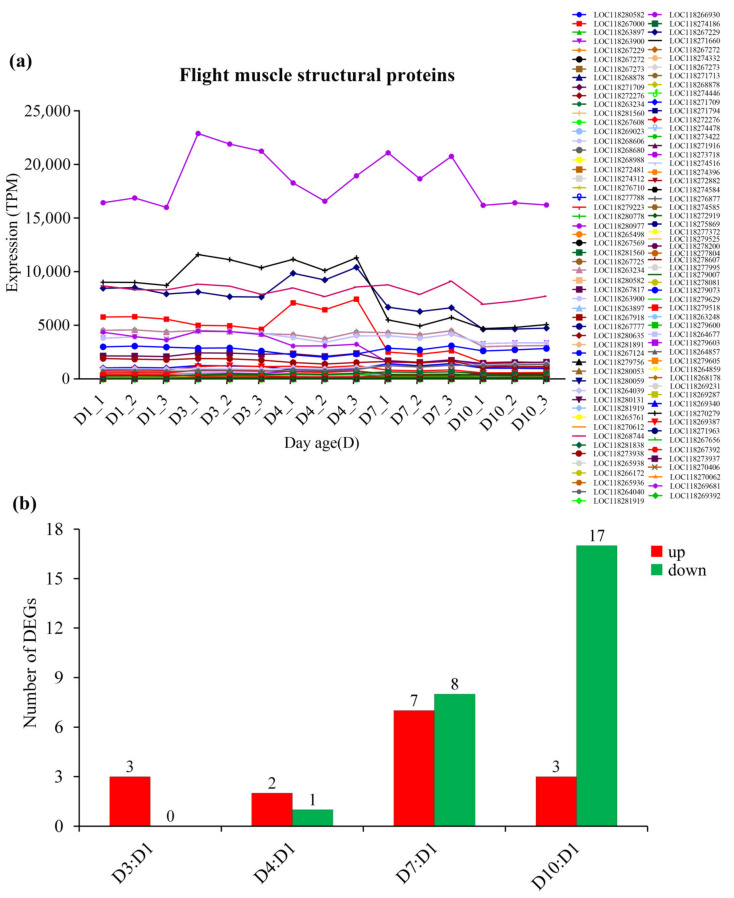
The expression of flight muscle structural proteins in *S. frugiperda* females at different ages. (**a**) Changes in gene expression with age; (**b**) DEGs of D3:D1, D4:D1, D7:D1 and D10:D1 groups, wherein red columns represent upregulated DEGs and green columns represent downregulated DEGs.

**Figure 10 insects-13-00936-f010:**
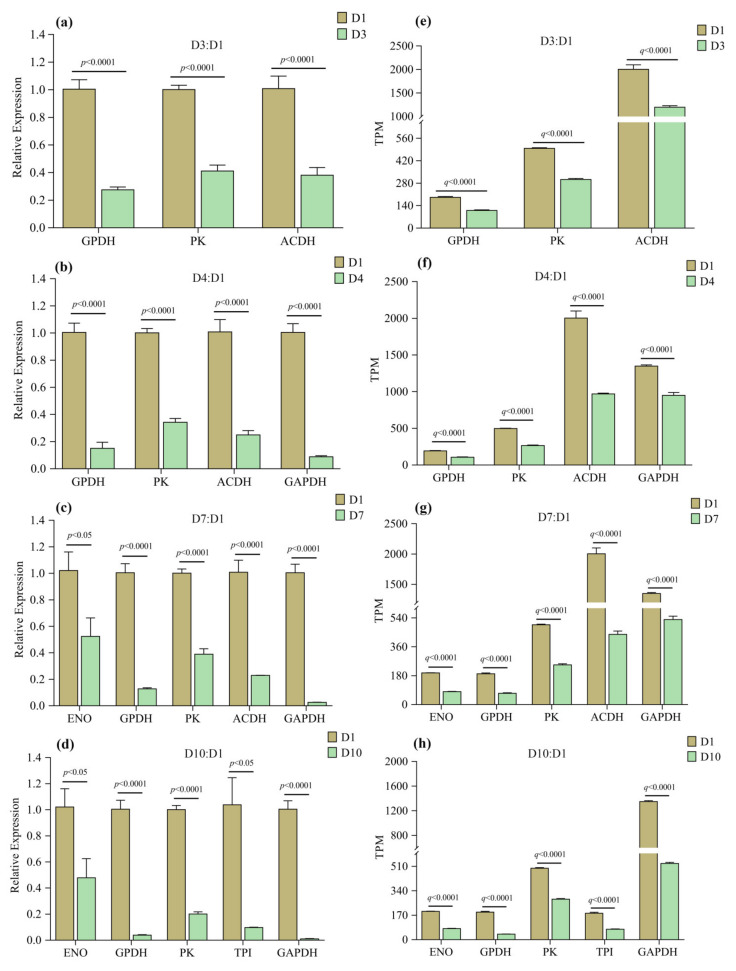
Transcriptome validation for DEGs by qPCR. (**a**–**d**) are relative expression levels of target genes measured by qPCR, and (**e**–**h**) are their expression levels measured by RNAseq. Error bars were standard error (SE).

**Figure 11 insects-13-00936-f011:**
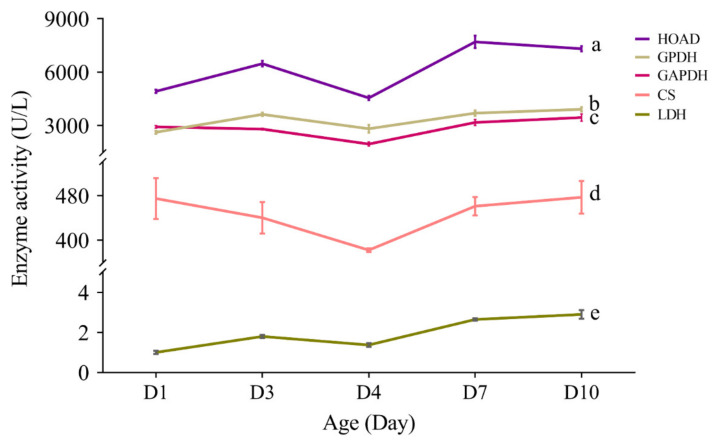
Activities of the five metabolic enzymes (GPDH, GAPDH, LDH, HOAD and CS) in the flight muscle of *S. frugiperda* females at different ages. Lines with different letters are significantly different (*p* < 0.05). Error bars were ± SE.

**Figure 12 insects-13-00936-f012:**
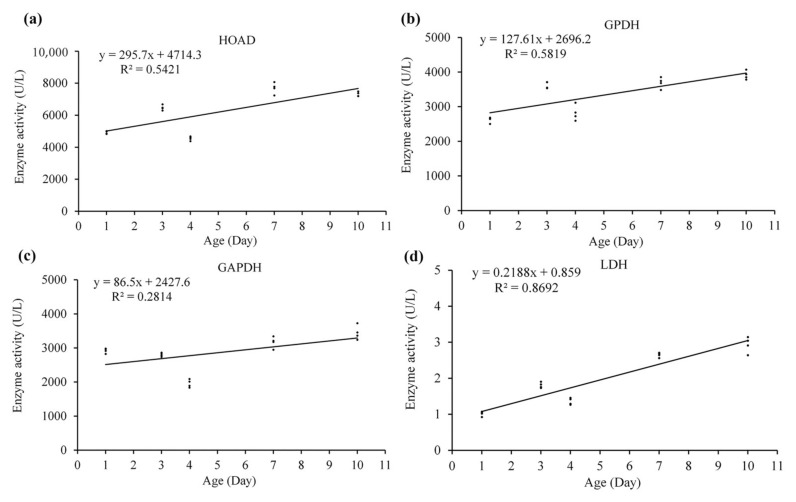
Linear regression between enzyme activity and age in *S. frugiperda* females. (**a**–**d**) are linear regression of HOAD, GPDH, GAPDH and LDH with age, respectively.

**Table 1 insects-13-00936-t001:** Enzymes and metabolites (full name and abbreviations) of flight energy metabolism in *S. frugiperda*.

Gene ID *	Gene Name	Abbreviation	Function	Reference
The glycolysis pathway				
LOC118280632	Trehalose	TRE	Catalyzing the conversion of trehalose to glucose.	[34]
LOC118276330	Hexokinase	HK	Catalyzes the conversion of glucose to glucose 6-phosphate (G6P).	[35]
LOC118265182	Glycogen phosphorylase	GP	Phosphorylation of glycogen to glucose 1-phosphate (G1P).	[36]
——	Glucose phosphate mutase	PGM	Catalytic conversion of G1P to G6P.	[9]
LOC118277306	Glucose-6-phosphate isomerase	GPI	Rearrange G6P into 6-phosphate-fructose (F6P).	[37]
LOC118270306	Phosphofructose kinase	PFK	Catalyzing F6P to produce fructose 1,6-diphosphate (FDP).	[38]
LOC118275099	Fructose-bisphosphate aldolase	FBA	FDP was decomposed into glyceraldehyde 3-phosphate (GAP) and dihydroxyacetone phosphate (DHAP).	[39]
LOC118268149	Glycerol-3-phosphate dehydrogenase	GPDH	Catalytic conversion between DHAP and glycerol 3-phosphate (G3P).	[15]
LOC118263360	Triose phosphate isomerase	TPI	Catalytic conversion between GAP and DHAP.	[40]
LOC118271716	Glyceraldehyde-3-phosphate dehydrogenase	GAPDH	Catalytic conversion of GAP to 1,3-diphosphoglyceride.	[15]
LOC118262502	Phosphoglycerate kinase	PGK	Catalyzes the conversion of 1,3-diphosphoglyceride to 3-phosphoglycerate.	[41]
LOC118274617	Phosphoglycerate mutase	PGAM	Catalytic conversion of 3-phosphoglyceric acid to 2-phosphoglyceric acid.	[42]
LOC118268804	Enolase	ENO	Catalytic conversion of 2-phosphoglyceric acid to phosphoenolpyruvate.	[41]
LOC118281561	Pyruvate kinase	PK	Catalyzes phosphoenolpyruvate to transfer high-energy phosphate groups to ADP to produce ATP and pyruvate (PYR).	[43]
LOC118272480	Lactate dehydrogenas	LDH	Catalyze the reduction and oxidation reaction between PYR and lactic acid (LA).	[44]
The fat oxidation pathway				
LOC118266592	Diacylglycerol lipase	DGL	Catalyze the hydrolysis of Diacylglycerol lipid (DG-LP) to release free fatty acids (FFA) and monoacylglycerol.	[45]
——	Diglyceride lipase	DL	Catalyzing the conversion of diglycerides to FFA.	[36]
LOC118262085	Acetyl CoA synthase	ACS	Catalytic synthesis of fatty acyl CoA (FA-CoA) from FFA.	[46]
——	Carnitine palmitoyl transferase	CPT	Synthesis of lipoacyl carnitine catalyzed by lipoacyl coenzyme A and carnitine.	[36]
LOC118273677	Carnitine O-acetyltransferase	CAT	Transfer of lipoacyl carnitine into mitochondrial matrix.	[36]
LOC118274443	Acyl-CoA dehydrogenase	ACDH	Make (C_n_) acyl coenzyme A in α and β One hydrogen is removed from each carbon atom to form β-enolyl-COA.	[47]
LOC118271967	Enoyl-CoA hydratase	ECH	Catalyzing trans alkenylacyl coenzyme A to produce β-hydroxyacyl COA.	[47]
LOC118272083	3-hydroxyacyl-CoA Dehydrogenase	HOAD	Catalyzing 3-hydroxyacyl coenzyme A to produce β-ketoacyl-COA.	[36]
LOC118265763	Thiolase	THI	Catalyzing the formation of (C_n-2_) fatty acyl CoA from β-ketoacyl-COA.	[47]
The proline oxidation pathway				
LOC118274501	Proline dehydrogenase	ProDH	The first step of catalyzing the catabolism of proline is to oxidize proline to pyrrole-5-carboxylic acid.	[48]
LOC118262586	Pyrroline-5-carboxylate dehydrogenase	P5CDH	Glutamic acid oxide- γ- Glutamic acid from semialdehyde.	[36]
LOC118268408	Alanine aminotransferase	ALT	Catalyzing the production of glutamate and alanine α-Ketoglutarate.	[36]
LOC118263189	Aspartate aminotransferase	AST	Catalyzing the production of glutamic acid and oxaloacetic acid α-Ketoglutarate.	[48]
Tricarboxylic acid (TCA) cycle				
LOC118269912	Pyruvate dehydrogenase	PDH	Catalytic oxidative dehydrogenation of PYR to acetyl-CoA.	[49]
LOC118262833	Citrate synthase	CS	The key enzyme at the entrance of the tricarboxylic acid cycle catalyzes the synthesis of citrate from oxaloacetate and acetyl CoA.	[50]
LOC118264042	Aconitase	ACO	Catalytic citrate acid to isocitrate.	[51]
LOC118267252	Isocitrate dehydrogenase	IDH	Catalytic oxidative dehydrogenation of isocitrate α-Ketoglutarate, the main factor controlling circulation.	[52]
——	α- Ketoglutarate dehydrogenase	α-KGDHC	Catalysis α-Ketoglutarate to succinyl CoA.	[53]
——	Succinyl COA synthase	SCS	It is a TCA cycle enzyme that catalyzes succinyl CoA to produce succinate.	[54]
LOC118271964	Succinate dehydrogenase	SDH	It is a key enzyme involved in the tricarboxylic acid cycle in mitochondria and catalyzes succinate to fumarate.	[55]
——	Fumarase	Fum	Catalyze the reversible hydration reaction between fumarate acid and malate.	[56]
LOC118269324	Malate dehydrogenase	MDH	Conversion between malate and oxaloacetate.	[57]
LOC118267676	Pyruvate carboxylase	PC	Catalyzing the carboxylation of H_2_CO_3_ and PYR to form oxaloacetic acid.	[58]

* Genes without IDs indicate they are present in insects but not found in the transcripts of *S. frugiperda.*

**Table 2 insects-13-00936-t002:** Flight muscle structural protein-encoding genes of *S. frugiperda*.

Protein	Gene ID	Function	Reference
Twitchin	LOC118280582 (twitchin-like)	Interacts with F-actin.	[59]
Tropomodulin	LOC118281560 (tropomodulin-1-like)	Terminal capping protein of actin.	[60]
Flightin	LOC118267000 (flightin-like)	A myosin-binding protein that maintains sarcomere stability.	[61]
Troponin	LOC118263234 (troponin T); LOC118263900 (troponin I); LOC118266930 (troponin C)	A sensor that receives Ga^2+^ to regulate muscle contraction; consists of three subunits, TnT, TnI and TnC.	[62]
Myosin	A total of 26 transcripts were identified. Such as LOC118278200 (myosin-11-like); LOC118278607 (myosin-14-like); LOC118279007 (myosin-7B-like); LOC118263248 (myosin-M heavy chain-like); LOC118264677 (myosin heavy chain); LOC118268178 (myosin heavy chain 95F-like); LOC118271963 (myosin-10-like)	A superfamily of molecular motors that move along the track of actin filaments, accounting for 60% of the total protein of myofibrils. Plays a role in muscle contraction.	[63]
Paramyosin	LOC118267608 (paramyosin-like); LOC118269023 (paramyosin, long form-like)	Play roles in myoblast fusion, myofibril assembly, and muscle contraction.	[60]
Connectin, Titin	LOC118268606 (titin-like, transcript variant X4); LOC118268680 (titin-like, transcript variant X1); LOC118268988 (titin-like, transcript variant X2) LOC118274312 (titin-like); LOC118267569 (connectin-like); LOC118267725 (connectin-like, transcript variant X1)	Intertwined with thick and thin filaments, making muscles elastic.	[60]
Tropomyosin	LOC118273937 (tropomyosin-1, transcript variant X1); LOC118273938 (tropomyosin-2, transcript variant X1)	Regulates myofibrils function by interacting with actin.	[64]
Actin	A total of 41 transcripts were identified. Such as LOC118279073 (actin, muscl); LOC118279603 (actin-like); LOC118279605 (actin, alpha skeletal muscle-like); LOC118280131 (actin-like protein 9A); LOC118264039 (actin-like protein 15A); LOC118264040 (actin-like protein 16A); LOC118268744 (actin-like protein 24A)	Play roles in muscle movement, accounting for 20% of the total protein of muscles. Divided into α, β and γ-actin three categories.	[65]

**Table 3 insects-13-00936-t003:** The activity ratio of GAPDH:HOAD in *S. frugiperda* at different ages.

Age (Day)	GAPDH:HOAD
D1	0.594
D3	0.433
D4	0.430
D7	0.412
D10	0.471

## Data Availability

The transcriptome raw reads have been deposited to the NCBI SRA database, the accession numbers are SRR20084632-SRR20084646. Other data generated or analyzed during this study are included in this article and its Appendix A.

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
