# Peer review of "The Transcription of Flight Energy Metabolism Enzymes Declined with Aging While Enzyme Activity Increased in the Long-Distance Migratory Moth, Spodoptera frugiperda"

_insects, 2022, doi:10.3390/insects13100936_

Round 1

Reviewer 1 Report

Fu et al. in this work investigated the transcriptional profiles and analyzed the primary metabolic enzyme activities in the migratory moth, Spodoptera frugiperda. RNA-seq showed that expression of the metabolic enzyme genes decreased with age, while the enzyme activities increased with age. These results look a little bit contradictory, but the authors argued that the downregulated metabolic gene expression indicates the stable enzymes at the protein levels. Also, they speculated lipids are the principal fuel in flight in this insect. Overall, this work establishes a transcriptomic data resource of Spodoptera frugiperda and provides some valuable information for studying insect flight. The experiment design, method and material description, data analysis, and writing are fine, so I just have one major and a few minor concerns.

 Major comment

The authors argued that lipid is the primary fuel of this species in flight, but they did not provide any physiological data. So, my question is if they could examine the profiles of total lipids or some form of fat with age. This may be a piece of easy but essential evidence for their speculation.

Minor comments

1.        Line 93, metabolic enzymes? Specify the key enzymes.

2.        Line 169, what kind of samples? What tissues? And is any proteinase inhibitor added into the lysis buffer? Please make them clear.

3.        Line 173, please add the catalogue number for each ELISA kit.

4.        Figure 10, I would like to suggest the authors also show the RNA-seq data of the selected enzyme genes nearby, allowing people to directly compare the qPCR and RNA-seq results.

5.        Line 331, what does the GAPDH:HOAD ratio mean? The authors should explain what they wanted to illustrate. I see they mentioned this in the Discussion section, but I believe it will be better also to state their intention here.

Author Response

Comments and Suggestions for Authors

Fu et al. in this work investigated the transcriptional profiles and analyzed the primary metabolic enzyme activities in the migratory moth, Spodoptera frugiperda. RNA-seq showed that expression of the metabolic enzyme genes decreased with age, while the enzyme activities increased with age. These results look a little bit contradictory, but the authors argued that the downregulated metabolic gene expression indicates the stable enzymes at the protein levels. Also, they speculated lipids are the principal fuel in flight in this insect. Overall, this work establishes a transcriptomic data resource of Spodoptera frugiperda and provides some valuable information for studying insect flight. The experiment design, method and material description, data analysis, and writing are fine, so I just have one major and a few minor concerns.

 Our answer: We appreciate these constructive comments and suggestions.

 Major comment

The authors argued that lipid is the primary fuel of this species in flight, but they did not provide any physiological data. So, my question is if they could examine the profiles of total lipids or some form of fat with age. This may be a piece of easy but essential evidence for their speculation.

 Our answer: We agree that further physiological data will further improve the quality of this study. We did not do this is because: (1) many previous studies have demonstrated that GAPDH:HOAD activity ratio can be reliable parameter for the evaluation of carbohydrate/lipid use pattern, which has been addressed and discussed fully in the MS; (2) the direct evidence of lifetime flight fuel pattern may need much more complicated study, such as need to consider the flight, reproduction and aging, which thus need a thorough study and can be another good story; (3) the object of this study is the thorax, and the focus is its transcription and enzyme activity, while the lipid is mainly in the fat body (which can be used for survival, flight and reproduction) that in the abdomen, we thus did not consider it in the present study. However, we thank this good advice and will consider it in our future works.

Minor comments

  1. Line 93, metabolic enzymes? Specify the key enzymes.

Our answer: Thanks and revised.

  1. Line 169, what kind of samples? What tissues? And is any proteinase inhibitor added into the lysis buffer? Please make them clear.

Our answer: Agree and provided some information in the MS. The manufacturer said it did contain multiple protease inhibitors, but could not tell us what the exact reagents it contained due to commercial secret.

  1. Line 173, please add the catalogue number for each ELISA kit.

Our answer: Done. We also updated the manufacturer’s name as the previous one is the dealer’s name.

  1. Figure 10, I would like to suggest the authors also show the RNA-seq data of the selected enzyme genes nearby, allowing people to directly compare the qPCR and RNA-seq results.

Our answer: We agree and have now added the RNAseq results in the figure.

  1. Line 331, what does the GAPDH:HOAD ratio mean? The authors should explain what they wanted to illustrate. I see they mentioned this in the Discussion section, but I believe it will be better also to state their intention here.
    Our answer: Agree and done.

Reviewer 2 Report

The present manuscript provides an interesting story. The manuscript is very well conducted, with clear results and documented discussion and conclusion. It is well written in general, with some issues which need to be clarified/added/changed prior to the final decision. Detailed info can be found in the attached pdf file.

Major points:

1.      Authors should explain why they only used females in this study in the introduction.

2.      Please write clearly how enzymatic activities were checked. Please make sure whether the measuring wavelength is the same for all enzymes or not.

3.      What types of statistical analyses are used in the study? Please make a separate paragraph in the method section.

Minor points:

Minor points can be found in the attached pdf file.

Author Response

The present manuscript provides an interesting story. The manuscript is very well conducted, with clear results and documented discussion and conclusion. It is well written in general, with some issues which need to be clarified/added/changed prior to the final decision. Detailed info can be found in the attached pdf file.

Our answer: We are very grateful to the positive comments and revision suggestions to our MS.

Major points:

  1. Authors should explain why they only used females in this study in the introduction.

Our answer: Agree and added more information to the last paragraph of Introduction.

  1. Please write clearly how enzymatic activities were checked. Please make sure whether the measuring wavelength is the same for all enzymes or not.

Our answer: More detailed information was added to this part.

  1. What types of statistical analyses are used in the study? Please make a separate paragraph in the method section.

Our answer: We agree and have now provided a separate paragraph for the statistical methods used in the qPCR and enzyme activity test.

 Minor points:

Minor points can be found in the attached pdf file.

Our answer: We thank these revision comments and have now revised the MS accordingly.

Round 2

Reviewer 1 Report

I have no further comments. Thanks for the authors' response.